# An Overview of Coastline Extraction from Remote Sensing Data

**Xixuan Zhou [1], Jinyu Wang [1], Fengjie Zheng [2], Haoyu Wang [1] and Haitao Yang [2],***

[1] Department of Graduate Management, Space Engineering University, Beijing 101400, China; zhouxixuan18@mails.ucas.ac.cn (X.Z.); hangchen@alu.hit.edu.cn (J.W.); 2018205188@qdu.edu.cn (H.W.)

[2] Department of Aerospace Information, Space Engineering University, Beijing 101400, China; zhengfj@radi.ac.cn

[*] Correspondence: 13400416091@sjtu.edu.cn

**Abstract:** The coastal zone represents a unique interface between land and sea, and addressing the ecological crisis it faces is of global significance. One of the most fundamental and effective measures is to extract the coastline's location on a large scale, dynamically, and accurately. Remote sensing technology has been widely employed in coastline extraction due to its temporal, spatial, and sensor diversity advantages. Substantial progress has been made in coastline extraction with diversifying data types and information extraction methods. This paper focuses on discussing the research progress related to data sources and extraction methods for remote sensing-based coastline extraction. We summarize the suitability of data and some extraction algorithms for several specific coastline types, including rocky coastlines, sandy coastlines, muddy coastlines, biological coastlines, and artificial coastlines. We also discuss the significant challenges and prospects of coastline dataset construction, remotely sensed data selection, and the applicability of the extraction method. In particular, we propose the idea of extracting coastlines based on the coastline scene knowledge map (CSKG) semantic segmentation method. This review serves as a comprehensive reference for future development and research pertaining to coastal exploitation and management.

**Keywords:** coastline extraction; remote sensing; deep learning; remote sensing knowledge map





## 1. Introduction

Coastal areas worldwide are experiencing rapid development, indicating significant growth potential. For centuries, these areas have served as the main drivers of economic development in many countries [1]. For example, the highest GDP density in China is found in Shanghai, Guangdong, and other provinces along the coast of southeast China [2]. However, the impact of climate change and human activity has led to coastline erosion and poses a significant threat to the economic and personal safety of coastal areas [3,4]. How to deal with these threats has become a difficult problem in coastal area management [5]. It is also the basis for coastal and Marine resource management, coastal zone regional environmental monitoring, and sustainable development planning [6]. Coastlines are constantly changing as part of the natural process or due to human influence, and coastline dynamic monitoring has become the basis to solve the problem of coastal area management [7].

Accurate coastline extraction is the basis for coastline change monitoring and coastal area ecological environment change monitoring [8,9]. Traditional extraction provides two main methods to extract coastlines: field studies, including site beach profiling and high-resolution aerial datasets for analysis and artificial recognition and manual vectorization from aerial imagery using professional software such as ArcGIS 10.3 developed by ESRI [10]. Both traditional methods have common weaknesses in that the labor cost is high and the statistical value needs to be more accurate. Moreover, fieldwork has certain personnel safety issues [11,12].

Over time, remote sensing technology has evolved, remote sensing platforms have updated, and sensors have emerged one after another. Remote sensing data acquired

by different sensors enables researchers to overcome the limitations and shortcomings caused by geological landscapes and long-temporal [13]. Abundant airborne platforms are the first to be used for earth observation. Plenty of aerial imagery acquired by different techniques are used for coastline extraction [14]. Airborne remote sensing data have certain limitations in both time and space coverage, and the emergence of Earth observation (EO) systems has revolutionized how we observe the landscape [15]. Satellite-based imagery, including hyperspectral imaging (HSI), multispectral imaging (MSI) [16,17], SAR [18], and LiDAR [19,20], has been used for the visual analysis of water/land information [15]. With freely available data, the extraction of coastlines from satellite data has been updated with the newest technological developments in remote sensing satellites, such as NASA's Landsat (which revisits every 16 days) and Sentinel-2 (5-10 days of revisit time) launched by ESA. In addition to optical imagery, space-born SAR instruments including ALOS PALSAR-2 and RADARSAT-1 are used to detect coastlines [21]. For the moment, coastline detection, coastline extraction, water body classification, and continuous coastline monitoring have been realized by processing spectral images captured by abundant instruments.

Due to breakthroughs in aerospace, sensor, and communication technologies, humans are now in a time where remote sensing (RS) big data are prevalent. This has shifted researchers' focus towards exploring new image processing and analysis techniques. Image processing methods strive to improve the separation of spectra between water and land, image segmentation or classification, feature detection and extraction, and pixel unmixing of RS images [22]. These methods are conducted by classic statistical algorithms and band math [23]. Meanwhile, rapid advances in accessing high-resolution remote sensing data have led to an explosion of big data, providing new opportunities for data-driven discovery [24].

Along with the rapid development of artificial intelligence (AI), the great success of machine learning (ML) has attracted widespread attention [25]. ML is a data-driven AI that can self-learn from sample data [26]. It enables classification and prediction by analyzing the relationship between the input spectrum and the desired output and can be used for complex tasks such as land cover classification [27], crop classification [28], and target recognition [29]. In the literature, by leveraging powerful computational capabilities and multi-dimensional auxiliary information, ML enables the extraction of more accurate coastlines from large-scale or long-time series of remotely sensed big data [30]. Applying ML to big data analysis and mining as a key technology helps researchers deal with coastline extraction problems and digital coastline management in complex situations. However, due to the highly dynamic nature of the coastline, the influence of tides and seawater erosion, it is still a challenge to accurately locate the changes in surface reflection characteristics [31].Given the increasing frequency and effectiveness of ML and DL methods in remote sensing, it is necessary to summarize the relevant applications of coastline extraction.

Compared with previous literature reviews, this paper focuses on various types of remote sensing images and methods for coastline extraction. This paper offers a comprehensive review of the application of a remote sensing index, thresholding methods, and machine learning methods in coastline extraction, while we demonstrate the characteristic of different types of coastlines and corresponding extraction methods. Firstly, we review the key attributes of popular RS data used for coastline extraction and obtained by drones and satellites in recent studies. Secondly, we discuss the features of different coastline types, including bedrock coastline, beaches, artificial coastline, silt coastline, and biological coastline. Thirdly, we investigate the evolution of the most popular and recently used methods for the automatic extraction of coastlines. We classify methods into four groups: water indices, edge detection, thresholding methods, and machine learning. Moreover, we summarize the extraction methods of different types of coastlines. Finally, we discuss present gaps and challenges in the science of coastline detection and extraction. This paper discusses on the data perspective and coastline perspective, providing a discussion of the applicability of different methods to different data and different categories of coastline

identification, providing new ideas of concern for method selection and the creation of feature datasets.

## 2. Materials and Methods

We begin by categorizing and grouping different coastline types and extraction methods, and then introduce the literature of ML methods related to coastline extraction. This paper reviews the extraction of coastlines using RS data processing and machine learning means. The main keywords used in searching the literature on the Web of Science (WOS) include "machine learning", "remote sensing", and "coastline extraction", which were identified through consultation with experts and based on previous research.

We graphed the number of publications in the last decades using various keywords from WOS. Figure 1a shows the number of papers using the keywords "remote sensing" and "coastline" from WOS. Figure 1b shows the search result based on keywords including "remote sensing" and "coastline extraction" or "coastline definition". Figure 1c shows the search result based on keywords including "machine learning", "remote sensing", and "coastline extraction" or "coastline definition".

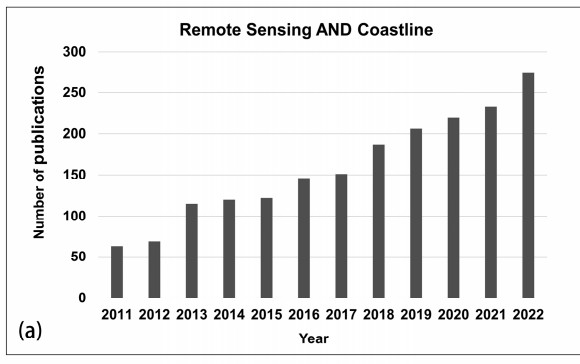

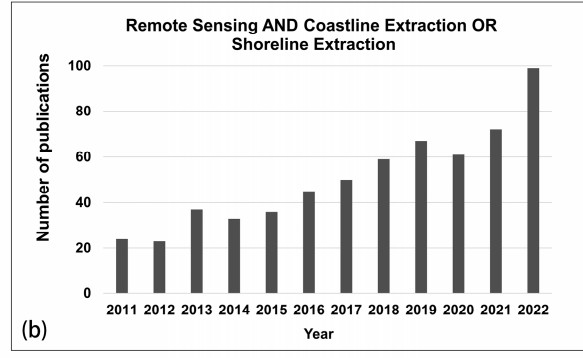

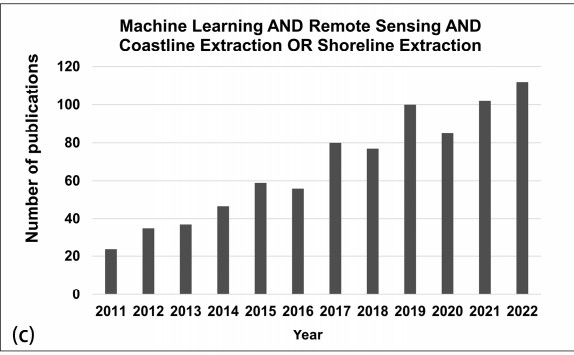

**Figure 1.** Number of publications in the last decade based on the WOS research publication database and obtained by searching keywords: (**a**) "remote sensing" and "coastline", (**b**) "remote sensing" and "coastline extraction" or "shoreline extraction", and (**c**) "machine learning" and "remote sensing" and "coastline extraction" or "shoreline extraction".

Based on the data trend, researchers are increasingly focusing on coastlines. Although the number of publications in Figure 1b,c is slightly lower compared to Figure 1a, they also exhibit a rising trajectory, indicating the growing interest among researchers towards utilizing machine learning for coastline extraction. Comparing Figure 1b,c, it is not difficult to find that machine learning methods take up a large proportion of the research on coastline extraction in the past ten years.

## 3. Available Data Sources for Remote Sensing Coastline Extraction

Remote sensing observation can be reduced by using sensors to obtain the electromagnetic wave information of the landscape object and the characteristics, changes, or trends of the observed object can be derived from electromagnetic wave information to achieve the purpose of earth observation. This paper summarizes commonly used remote sensing data characteristics for coastline extraction based on different platforms and sensors.

### 3.1. Satellite Remote Sensing Data

Generally, remote sensing can be divided into active and passive remote sensing, which are defined based on the source of energy used to collect data [24]. Passive systems rely on natural radiation sources (such as the sun) emitted or reflected by the object. The active system receives and records the reflection of the wave it emits towards the target. Optical and radar data, categorized as passive and active remote sensing data, respectively, constitute the most important part of data applied in coastline extraction. We review popular optical and radar sensors in Section 3.1.1 and Section 3.1.2, respectively.

### 3.1.1. Optical Data

The U.S. LANDSAT family of satellites is jointly managed by NASA and the U.S. Geological Survey (USGS) [32]. Since 1972, the LANDSAT series of satellites have been launched successively. The series of Landsat satellites have provided temporal data for coastline extraction and change monitoring over the last decades.

Among these satellites, Landsat-5, Landsat-7, and Landsat-8 have gained more attention from researchers in coastline extraction [1,16,32–35]. The near-infrared and visible wavelength bands of Landsat satellites were applied to determine the land and water surfaces along the coastline [16]. Landsat-5 is the fifth satellite in the U.S. LANDSAT family, launched in March 1984 as an optical Earth observation satellite with a payload of a Thematic Cartographer (TM) and Multispectral Imager (MSS). The images obtained by Landsat-5 are commonly used for land–sea separation and coastline extraction due to their advantages of multi-spectral and spatial resolution [16]. Landsat-7 has a payload of an Enhanced Thematic Mapper Plus (ETM+), which provides high-resolution data resulting from an additional band with a resolution of 15 m, and was launched in April 1997. Landsat-8, launched in February 2013, is the successor to the U.S. Landsat fleet. The Landsat-8 is distinguished by an Operational Land Imager (OLI) that encompasses a spectrum of wavelengths ranging from infrared to visible light, alongside a Thermal Infrared Sensor (TIRS) [36]. Compared with the ETM+ sensor of Landsat-7, OLI has a blue band (0.433–0.453 μm) and a near-infrared band (Band 9; 1.360–1.390 μm), and the blue band is mainly used for coastal zone observation [36]. Among different Landsat satellites, the combination of Landast-5 and Landsat-7 is prevalent [5].

Worldview-2 (wv-2) is affiliated with the Worldview satellite family launched by DigitalGlobe Inc. (Westminster, CO, USA) in October 2009. WV-2 is capable of capturing multispectral imagery across eight distinct bands [3]. DigitalGlobe Inc. also owns IKONOS, which was launched in 1999. Although IKONOS has decommissioned, the capability for stereo mapping, as well as its contribution to coastline extraction, is still worthy of recognition.

Sentinel-2 is equipped with a multi-spectral instrument (MSI) that can cover 13 spectral bands from 442 nm up to 2202 nm with different resolutions: 10 m (Central Wavelength (CWL) at 490, 560, 665, and 842 nm with bandwidths of 65, 35, 30, and 115 nm, respectively),

20 m (CWL at 705, 740, 783, 865, 1610, and 2190 nm with bandwidths of 15, 15, 20, 20, 90, and 180 nm, respectively), and 60 m (CWL at 443, 940, and 1375 nm with bandwidths of 20, 20, and 30 nm, respectively) [37]. Similar to WV-2, Sentinel-2 also has a coastal band that monitors near-shore bodies of water. In previous studies, the combination of Sentinel-2 and Landsat has been widely used for coastline extraction [10]. In light of the present situation, researchers are granted complimentary access to a diverse range of data from various data platforms, including Landsat and Sentinel-2. Google Earth Engine (GEE) offers a robust platform for researchers to acquire and process large-scale data efficiently [38].

In addition to the satellite data mentioned, ribosome data are used less frequently. SPOT (Satellite Pour l'Observation de la Terre) is an earth observation satellite system developed by the French Space Research Centre (CNES) [39]. The SPOT satellite system consists of a family of satellites and ground-based systems for satellite control, data processing, and distribution [7,40]. The IRS series supports the development of agriculture, water resources, forests and ecology, geology, water conservancy facilities, and coastline management in India, launched by the Indian Space Research Centre (ISRO). The composite map derived from the data of two MSI satellites, IRS-P6 and IRS-2, was successfully utilized for coastline extraction in this satellite series [41].

In an optical remote sensing image, an object's feature is reflected in the radiation difference of its electromagnetic wave. This reflection characteristic is utilized for discriminating between ocean and land, extracting coastlines. To extract coastlines using optical images, it is important to consider the features of different bands that better adapt to the coastline image features in different geographical environments.

### 3.1.2. Radar Data

The utilization of optical data in feature detection and extraction based on visible channels has gained extensive recognition [21]. Where features are partially impeded by environments (vegetation, clouds, etc.), the application of Radar data can overcome these interferences [34]. The RADARSAT program encompasses RADARSAT-1 (1995) and RADARSAT-2 (2007), which were launched by the Canadian Space Agency (CSA) [42]. These satellites are equipped with a synthetic aperture radar (SAR) instrument that is extensively utilized for oceanic, sea ice, and coastline monitoring purposes [43]. The ALOS satellite is equipped with three payloads, namely the Pan-Color Remote Sensing Stereogram (PRISM) for precise digital elevation mapping, the Advanced Visible and Near Infrared Radiometer 2 (AVNIR-2) for accurate land observation, and the Phased Array L-Synthetic Aperture Radar (PALSAR) for continuous all-day land observation regardless of weather conditions [44,45]. COSMO-SkyMed carries high-resolution radar satellites equipped with X-band SARs that can be used for coastline extraction [46]. In addition to those mentioned above, Sentinel-1 is also commonly used for coastline extraction of SAR data to extract reliable data for multiple coastlines.

Radar data exhibit drawbacks such as limited spatial resolution and reduced interpretability due to speckle noise; however, their unparalleled quasi-all-weather capability for continuous observation makes them an indispensable component of Earth observation sensors [21]. Moreover, land areas are generally light and sea areas dark, facilitating coastline extraction in SAR images. Especially when extracting sandy coastline and silty coastline, SAR images make it easier to extract accurate coastlines due to their sensitivity to water content.

### 3.2. Non-Satellite Remote Sensing Data

Unmanned Aerial Vehicle (UAVs) remote sensing is another choice for coastal research in recent years [47]. Several past studies demonstrated that UAVs equipped with sensors are more effective in small-scale coastline management since UAV surveys are capable of collecting timely and higher spatial resolution data [48,49]. At present, the most commonly used UAV remote sensing technology is airborne LiDAR to quantify coastline features [50].

In addition to LiDAR instruments, optical payloads such as traditional RGB cameras and hyperspectral sensors are also indispensable [20,49].

Among these sensors, LiDAR overperforms others resulting from it and can directly derive point clouds. Namely, it can accurately delineate coastlines from image-based systems utilizing Structure from Motion (SfM) analysis, without identifying corresponding features in overlapping scans [20,51,52]. Utilizing LiDAR multi-return point data, the incorporation of LiDAR intensity values enables the identification of rocks, particularly susceptible sections of revetment rocks that may be impacted by rising sea levels [53].

The use of LiDAR-equipped UAVs for coastline surveying to obtain timely, high-resolution, and site-specific coastline data is suitable for extracting precise rocky coastlines on a small scale, as well as quantifying highly dynamic coastlines over short timescales [49,53]. Especially in bedrock coastline extraction, LiDAR intensity values overperformed in discerning rock identification and coastline extraction.

## 4. Coastline Types and Indicators

### 4.1. Coastline Types

Coastlines are affected by the local coastal zone's topography, landform, and development degree. Thus, coastlines show different types and various geometrical morphologies in complex offshore environments [50]. Classifying them according to their coastal substrate characteristics and spatial morphology, these coastlines are subdivided into 16 categories including exposed rocky shores, exposed rocky platforms, fine-grained sand beaches, coarse-grained sand beaches, mixed sand and gravel beaches, gravel beaches, riprap structures, exposed tidal flats, sheltered rocky structures, sheltered man-made rocky structures, sheltered tidal flats, salt to brackish marshes, freshwater marshes, swamps, and mangroves [52]. In the introduction of methods for extracting different types of coastlines, the above types are reclassified into five classes, described in Table 1.

**Table 1.** Description of coastline types.

| Types | Interpretation Signs | Location |
| --- | --- | --- |
| Rocky coastlines | Large curvature, serrated shape | Use the land–water boundary as the coastline |
| Sandy coastlines | Relatively straight and the beach is striped, often with a beach ridge stacked up | Use the beach ridge as the coastline |
| Silty coastlines | Relatively straight, gentle slope, with large differences in vegetation density between the two sides of the intertidal zone | Use the obvious boundary between the tidal beach and the salt-tolerant vegetation as the coastline |
| Biological coastlines | The tidal ditch is obvious, with mangroves and other plants growing | Use the vegetation density difference as the coastline |
| Artificial coastlines | Straight direction, regular shape, and steep slope | Use the edge of artificial structures as the coastline |

Rocky coastlines are the most prevalent of these coastlines, with nearly 75% of the world's ocean coastlines having sea cliffs or cliffs [54,55]. Figure 2 shows a photo of the rocky shoreline located in Jekyll Island State Park, USA. In previous coastline extraction studies, the objects extracted were also mostly rocky coastlines. These coastlines are usually characterized by alternating rocky headlands and recessed bays with a large degree of curvature, with the headlands protruding into the sea and the bays penetrating deep into the land [54].

The intertidal substrate of the sandy coastline is generally dominated by sand and gravel, which refers to the relatively flat coastline formed by sand, gravel, and other sedimentary materials under the long-term action of waves. In areas inaccessible to waves, there is no accumulation of sand. Therefore, the demarcation between the sandy beach and areas without sand coverage can be regarded as the precise location of the sandy coastline [56]. The sandy beach is brighter than the non-sandy cover as seen in the images, and the sandy coastline is quite obvious [57]. Figure 3 shows an optical image of the sandy shoreline obtained by Sentinel-2 Satellite.

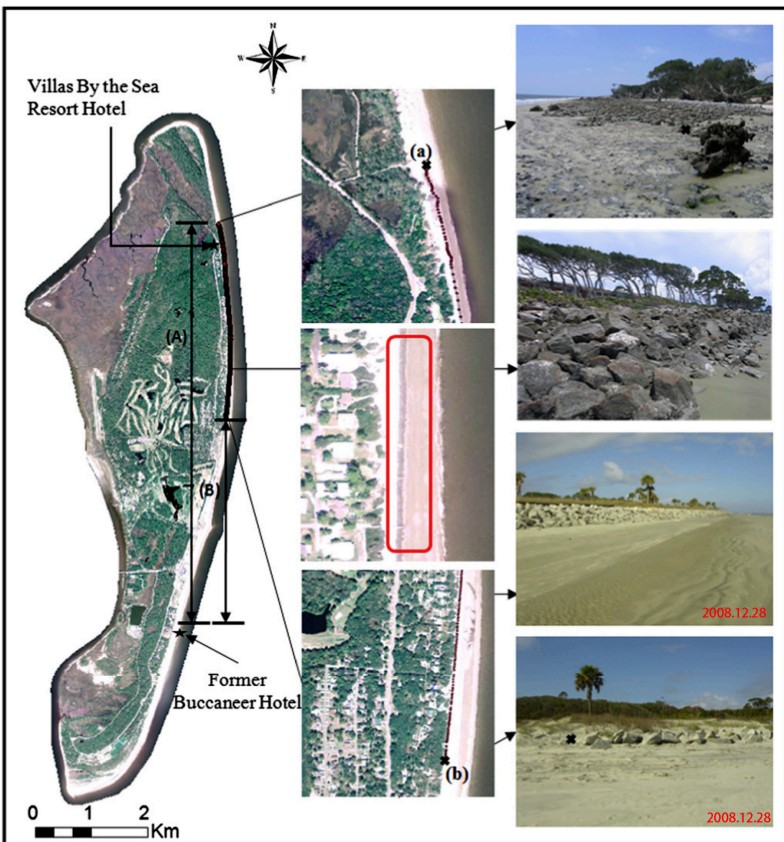

**Figure 2.** A schematic of a rocky coastline. The rocky shoreline is located in Jekyll Island State Park, USA. Currently exposed revetment rocks are shown in beach segments (A) and (B). (B) Part of the revetment rock shown in (A) has been buried by sand, only approximately 3.5 km of rocks from (b) to (a) are still exposed [53].

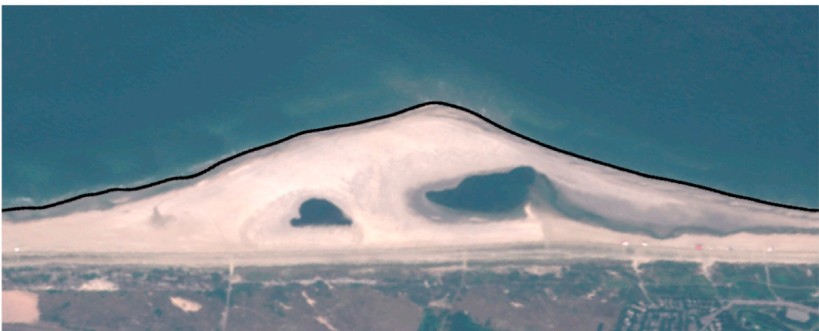

**Figure 3.** Satellite image of the Shamada research site obtained by the Sentinel-2 satellite. The black line shows the sandy coastline [57].

The intertidal substrate of silty coastlines is powdered sandy silt, an open coastline formed by sediment with a grain size of 0.01~0.05 mm under the dynamic action of tide and runoff over a long period of time. A silty coastline is flat and open, and the beach is several kilometers or even more than 10 km wide, which is the main distribution area of coastal mudflat wetlands. There is much salt-tolerant vegetation that typically thrives on the landward side of the intertidal zone (the area where the ocean meets the land between high and low tides) [56]; thus, the coastline of the silty coast can be defined as the obvious boundary between the tidal beach and the salt-tolerant vegetation. Figure 4 shows this boundary clearly.

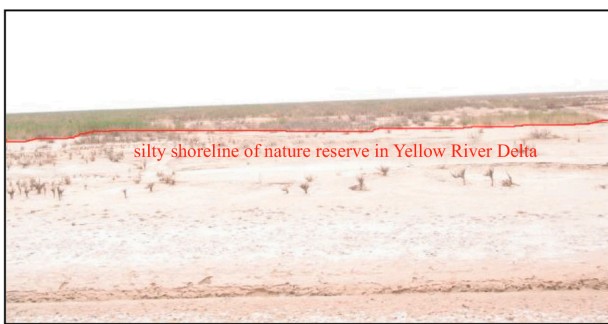

**Figure 4.** Photo of silty coast [56].

Biological coastlines are mostly located in tropical areas at low latitudes, mainly mangrove coastlines, coral reef coastlines, shell dike coastlines, and so on. Mangrove coastlines are areas where salt-tolerant and dense mangrove forests grow and are currently the most studied biological coastlines due to their sensitivity to saltwater intrusion [58]. Mangrove distribution can represent a change in the coastline. A previous study demonstrated that mangrove patches could be identified from remote sensing data based on color (medium-dark green), texture (cauliflower pattern), shape (dendritic perimeter), and location (close to the coastline) features [59]. Figure 5 shows the photo of biological coast.

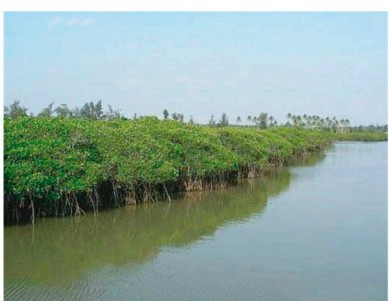

**Figure 5.** Photo of biological coast [56].

Artificial coastlines have artificial construction characteristics formed by constructing common artificial structures (rock armor and seawalls) and other means [9]. artificial coastlines are characterized by a straight direction, regular shape, and steep slope in spatial form. Compared with natural coastlines, artificial coastlines are stable and unaffected by tides and other factors and can be extracted from high-resolution satellite images in time series. Therefore, artificial coastlines are easier to extract than natural coastlines. Figure 6 shows a satellite image of artificial coast obtained by Sentinel-2 satellite.

According to the characteristics of different coastlines summarized above, it is necessary to select appropriate remote sensing images for coastline extraction. Optical data contain rich spectral information, which can distinguish water, land, and vegetation by different spectral responses. However, due to different shooting angles, shadow and stacking phenomena will occur in complex terrain areas, which will affect the determination of the coastline position. In SAR data, the water is dark and the land is bright, which helps to extract coastlines, and SAR data are unaffected by clouds and fog. However, the spatial resolution of SAR data is low, it is susceptible to the influence of speckle noise, and there are also shadows and overlapping phenomena. LiDAR point cloud data have strong penetration and are unaffected by shadows, but they cannot be operated on a large scale. For the extraction of rocky shorelines, LiDAR data will be more suitable than optical, especially without the impact of cliff shadows. In the extraction of sandy shorelines, coastlines can be extracted from optical data by exploiting the difference characteristics of spectral responses, and water content can also be distinguished by different backscattering coefficients in SAR data. The determination of the location of the silt coastline depends on the distinction between the intertidal zone and the halophyte zone. The obvious difference in spectral

response between vegetation and sand beach is the basis for the distinction, so optical data are the most appropriate data. Similarly, optical data are the most suitable for vegetation coastal zone extraction. When identifying artificial coastal zones, all three can be selected, and high spatial resolution data are the first choice.

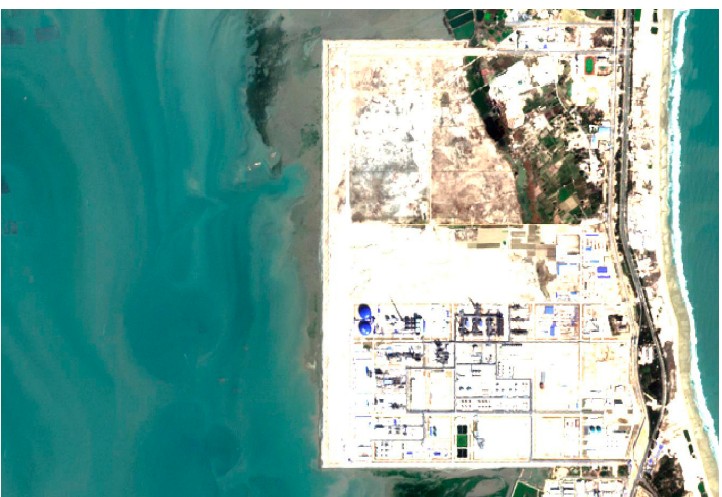

**Figure 6.** Satellite image of artificial coast image obtained by the Sentinel-2 satellite.

*4.2. Coastline Indicators*

Coastline indicators are visual features that represent the coastline position; different coastlines have different interpretation indicators [56]. These indicators can be divided into three groups: visually distinguishable coastal features, coastlines copied as benchmarks, and indicators based on image processing. Each user has a different definition of coastline indicators or has their own interpretation of the coastline indicators of a specific method, which will lead to different selections of coastline extraction methods and deviation of extraction results [60]. Consequently, determining appropriate coastline indicators, data, and extraction methods according to the coastline type is relatively accurate.

In satellite images, instantaneous water lines, wet/dry lines, vegetation limits, artificial limits, and morphological lines can be extracted as coastline indicators to discern coastal features. Most previous studies extracted instantaneous waterlines as a common coastline indicator. The most common coastline extraction technique for extracting instantaneous waterlines is remote sensing indices such as the Normalized Difference Water Index (NDWI) [58], modified NDWI (MNDWI) [10,17,61], and Automatic Water Extraction Index (AWEI) [35], which will be detailed in Section 5.1.

The wet/dry line represents a distinct boundary, demarcating the interface between saturated and unsaturated. Several publications have appeared in recent years documenting wet/dry boundaries which were be extracted from optical and SAR images by distinguishing and clustering pixels with different characteristics, such as soil moisture [62].

Remote sensing indices and machine learning are effective for vegetation lines and artificial boundaries. The vegetation line serves as a coastal indicator that demarcates the boundary between vegetation and water, which is determined through pixel value differentiation. Artificial limits such as man-made embankments are similar to vegetation limits that can be extracted by indeces and ML from multispectral images (MSI).

Morphological limits usually refer to the clear morphological lines of the shoreline, such as the tops and bottoms of cliffs [63]. In contrast to the aforementioned visually identifiable characteristics, morphological limits are difficult to extract from MSI precisely. In the case of a cliff, for example, the difference between the top and bottom of a vertical cliff is susceptible to shadows in MSI. LiDAR overperformed MSI in extracting these lines, resulting from it recognizing the dramatic change in elevation from top to bottom of a cliff.

The methods for the indicator extraction techniques in the above content will be described in more detail in Section 5.

## 5. Coastline Extraction Techniques

The present section provides a comprehensive overview of diverse techniques employed for coastline extraction from satellite imagery. In addition to elucidating the underlying theories and procedural steps associated with different methodologies, it also demonstrates the interrelationships between coastal indicators, extraction approaches, and utilized data. Figure 7 shows coastline extraction techniques has been used in the past.

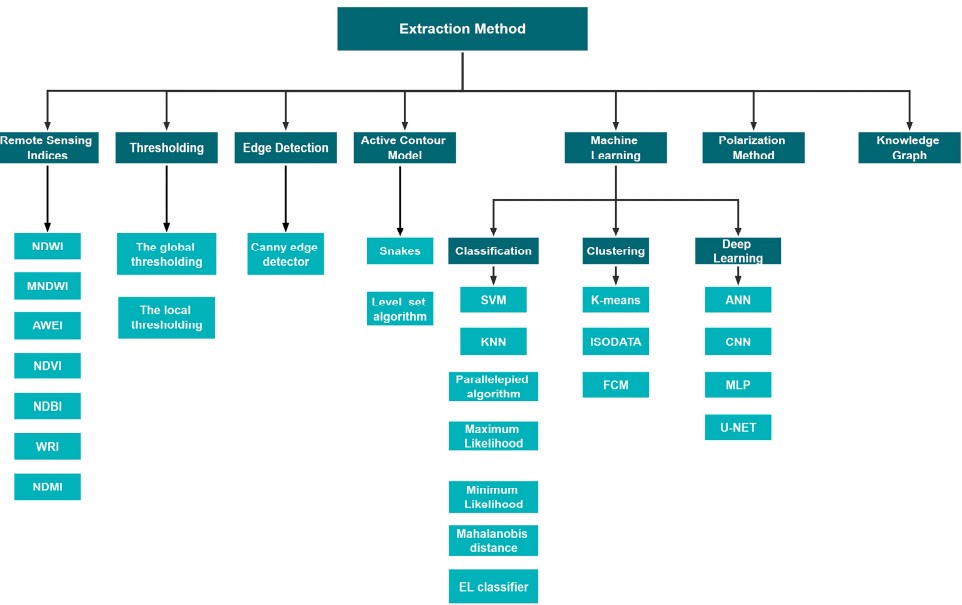

**Figure 7.** Methods that have been used in the past which have been applied to the extraction of the coastline (Table 2).

**Table 2.** Sample studies focused on extracting different types of coastlines in remote sensing data analysis.

| Coastline Types | Indicators | Extraction Technique |
|---|---|---|
| Rocky coastlines | Morphological limits | Using differences between LiDAR intensity values of multiple return points on revetment rocks and water to locate coastline position [53]. |
| | | A dual-polarization model takes full benefit of the PingPong model peculiarities were exploited to distinguish sea surface from land, then used image processing technique to extract coastlines [64]. |
| | | An integrated model of Convolutional Neural Network (CNN) and Object-Based Image Analysis (OBIA) was used to extract coastlines from remotely sensed images [65]. |
| | | Extracting coastline by an automatic approach for coastline detection from images which is based on parametric active contours(snakes) [66]. Select the applicable model through the supervised classification of ice, water, and rock. |
| Sandy coastlines | Morphological limits | Using the Sentinel-2 Water Edges Dataset (SWED), develop a convolutional neural network design based on U-Net for detecting coastline morphology [67]. |
| | | The coastline was extracted in two steps. The first step is using the Level Set Algorithm (LSA) to obtain the coarse boundary, then the contour is processed finely by LSA in a high-resolution image based on the coarse boundary [68]. |
| | | Improving the sea-land segmentation performance by modifying Standard U-Net, and developing an automatic coastline extraction framework to extract coastline from sea-land segmentation results [69]. |

**Table 2.** *Cont.*

| Coastline Types | Indicators | Extraction Technique |
|---|---|---|
| | | Identify the coastline using Pix4Dcapture, Pix4Dmapper, and ArcGIS 10.3 software to use the images captured by unmanned aerial vehicles [70]. |
| | | The fuzzy approach generated a classification SAR image to distinguish the coastal pixels from the land surface pixels. The classified map is converted to vector form, and the Douglas-Peucker regularisation algorithm is applied to remove the zigzag effects and reconstruct the boundary [18]. |
| | Instantaneous waterline | The waterline was extracted using the Normalized Difference Water Index (NDWI) with the Canny edge detection and thresholding used to create a binary image of land water [71]. |
| | | A robust extraction method using an artificial neural network (ANN). ANN uses the feedforward NN to classify the pixels of SAR imagery into two categories, land, and sea. The coastline location is then determined as a boundary of these two groups of classified pixels [72]. |
| | | An ensemble automatic shoreline segmentation system (WaterNet) based on deep learning architectures to obtain coastlines automatically [73]. |
| | | The classification of water on land employs two ensemble classifiers, namely a majority voting ensemble classifier utilizing random forest and support vector machine with RBF kernel, and another ensemble classifier combining multi-layer perceptron and k-nearest neighbor [74]. |
| | | The satellite images of the coastline were analyzed using edge detection filters, mainly Sobel and Canny [21]. |
| | Wet/dry line | The classified image was regrouped into two classes (land and sea) by ISODATA classification technique [74]. |
| | | Supervised edge detection is used on optical remote sensing data to map wet/dry indicators in the sandy part of beaches [62]. |
| Silty coastlines | Wet/dry line | An object-oriented multi-scale segmentation method is used for automated extraction and classification of coastlines from remote sensing imagery [75]. |
| | | The NDWI and Otsu thresholding converts the image into a binary image. The coastline is delineated using binarized images which are produced from a thresholding-based segmentation algorithm [5]. |
| Biological coastlines | Vegetation limits | The vegetation and non-vegetation parts of the mangrove were distinguished by binary classification method based on the NDVI map [76]. |
| Artificial coastlines | Artificial limits | Extract coastline of man-made construction areas from airborne lidar data. Determination of pre-extracted coastline elevation by distinguishing the echo intensity of water and land, then translate coastline point cloud and generate coastline [9]. |

Nowadays, many diverse techniques are used for coastline recognition and extraction from RS imagery, which can be grouped into three categories including indexing methods, edge detection, and classification. Indexing methods focus on two aspects: remote sensing indices and thresholding. As for edge detection approaches, the extraction of coastlines is approached as an edge detection problem for water bodies or oceans in the proposed method. From the perspective of image classification, these classification methods are mainly based on object-oriented and pixel-oriented classification [75]. RS image classification is mainly based on the spectral characteristics of features, which are multi-band measurements of the electromagnetic radiation of features, and these measurements can be used as the original feature variables for remote sensing image classification. Additionally, texture, shape, and other topographical features are integrated with spectral characteristics applied to the analysis. Coastline extraction is a complex problem that often requires a combination of methods, rather than a task that can be accomplished based on one method alone.

### 5.1. Remote Sensing Indices

Coastline extraction methods using multi-spectral image data captured by optical remote sensing satellites can be divided into single and multi-band methods. The basis of the single-band coastline extraction method is that the body of water has a higher light absorption in the infrared band, while vegetation and soil have a higher reflectance in the infrared band [57]. This method requires that the image must contain an infrared band. The multi-band method is mainly based on the absorption and reflectance of water and land in different bands to segment land and water [36]. This method reaps the maximum benefits from diverse sources of information, making the target's characteristics stronger and the segmentation more precise. However, due to a large amount of multi-spectral data, complex mathematical operations will increase the processing burden.

The remote sensing indices are formed through band combination and calculation to magnify the difference in reflection information between ocean and land. For instance, the NDWI is a band ratio technique that uses the Green and NIR bands to produce greyscale images to offer a positive outcome for the water features and a negative value for the non-water features [31,57,58]. Since water pixels identified using NDWI include false positive values, an MNDWI was developed to improve the accuracy of water pixel extraction [39,50,77]. In practical studies, the study area often covers a variety of landforms, including beaches, tidal flats, and bedrock. The AWEI was previously constructed to distinguish between land and water, and the index has been applied and validated in different coastal environments [37,50]. For biography coastline extraction, a vegetation line is an example of a coastline indicator representing vegetation limits, which can be identified by the Normalized Difference Vegetation Index (NDVI) [76,78]. Moreover, the Normalized Difference Building Index (NDBI) is appropriate for artificial structure coastline extraction owing to the ability of this index to identify artificial buildings. In addition to these indices commonly used in coastline extraction, others perform well in water pixel identification but have not yet been applied in coastline extraction, such as the Water Ration Index (WRI) and Normalized Difference Moisture Index (NDMI). We presented the formula for the calculation of these indices in Table 3.

**Table 3.** Remote sensing indices for coastline extraction.

| Index | Equation | Remark | Reference |
|---|---|---|---|
| Normalized Difference Water Index | $NDWI = (Green - NIR)/(Green + NIR)$ | The water pixel has a positive value | [79] |
| Modified Normalized Difference Water Index | $MNDWI = (Green - MIR)/(Green + MIR)$ | The water pixel has a positive value | [80] |
| Automated Water Extraction Index | $AWEI = 4(Green\text{-}MIR) - (0.25\ NIR + 2.75\ SWIR)$ | The water pixel has a positive value | [81] |
| Normalized Difference Vegetation Index | $NDVI = (NIR - Red)/(NIR + Red)$ | The vegetation pixel has a positive value | [82] |
| Normalized Difference Building Index | $NDBI = (MIR - NIR)/(MIR + NIR)$ | Artificial structure pixels have a positive value | [83] |
| Water Ration Index | $WRI = (Green + Red)/(NIR + MIR)$ | The water pixel has a positive value | [84] |
| Normalized Difference Moisture Index | $NDMI = (NIR - MIR)/(NIR + MIR)$ | The water pixel has a positive value | [85] |

When faced with different types of coastline objects, the choice of indices for identifying coastline features is still a debatable issue. Compared with single-band methods, remote sensing indices can better reduce the sensitivity of some bands to fog, inhibit sea area features, and enhance the contrast between foreground and background. However, remote sensing indices can only complete the sea–land segmentation, and it is necessary to combine other methods to achieve coastline extraction.

### 5.2. Thresholding

Sea-Land Segmentation (SLS) is employed to delineate the oceanic and terrestrial regions in remote sensing imagery, with the boundary pixels between these segmented areas serving as a reliable representation of the coastline. Thresholding is widely considered to be the usual and the most efficient method of SLS. The method is readily comprehensible, straightforward to implement, and highly efficacious. The thresholding process involves selecting appropriate spectral bands based on the spectral characteristics of water and land, to establish a model that can effectively divide an image into two main uniform regions: water and land [86,87]. In practice, the values of the remote sensing indices mentioned in Section 5.1 or single-band reflectance are treated as objects to be clustered. The demarcation of the coastline can be established by delineating the interface between land and water. Thresholding contains two broad categories applied in SLS: global threshold segmentation (histogram statistics, entropy method, and Otsu) and local threshold segmentation (adaptive thresholding).

The global thresholding method is to set a threshold, where pixel values greater than this value are 1, and those less than this value are 0. Among these algorithms, the Otsu algorithm is representative of the global threshold. Otsu is an efficient algorithm for the binarization of images proposed by Japanese scholar OTSU in 1979, which is an adaptive method of threshold determination and optimal segmentation in the sense of least squares [88]. The essence of Otsu lies in clustering, where the attainment of desired results is contingent upon achieving a balanced distribution of pixels across each class [16].

The local thresholding method is to find the threshold for each pixel, where greater than this value is 1 and less than this value is 0. Utilizing a single global threshold for the entire image to delineate the water/land boundary may lead to undetectable local coastline edges due to the heterogeneous contrast in image intensity, resulting in fragmented coastline edges within low-contrast regions of the image [89]. However, the local threshold method dynamically sets the threshold based on the local features of adjacent pixels [90]. At this point, the local thresholding method overperforms global thresholding. The Sauvola algorithm is the benchmark for local thresholding methods. The Sauvola algorithm takes a gray image as input, with the current pixel at its center. The threshold of each pixel is dynamically determined based on the local mean and standard deviation of grayscale values surrounding the current pixel. The approach of fitting a bimodal Gaussian curve for analyzing and determining local thresholds has also been employed in a previous study [70].

Thresholding methods can mine more spectral information, making them suitable for biological coastline and silty coastline extraction. Selecting appropriate thresholds helps to differentiate between vegetation and ocean or tidal beach areas based on remote sensing indices map. This method is easy to understand, simple to implement, and effective. However, excessive reliance on spectral information while disregarding the spatial characteristics of the image can lead to significant errors in boundary classification, thereby resulting in diminished accuracy in detecting transition zones [86]. In addition, the selection of the threshold value will also directly affect the accuracy, and it is necessary to determine the appropriate threshold value through accurate measurement experimental results before large-scale coastline extraction.

### 5.3. Edge Detection

Edge detection is widely used in coastline extraction from satellite images, which is treated as an edge detection problem. Edge refers to the spatial domain where significant changes in gray, color, or texture features occur rapidly within a remote sensing image [91]. They usually occur at the boundary of two different regions: water and land. Many edge detection methods have been successfully applied to coastal detection, including the Canny algorithm [92,93], Sobel [21,66], Robert, and Prewitt. Here, we take contour detection including Snakes and level set algorithms as a part of edge detection. Most of the time, we

can detect the edge first, and then further process the detected edge to get the contour of the target.

Canny algorithm is a technique for extracting edges between different visual objects [93]. The extracting result has good continuity and no breakpoints in principle. Compared with Sobel, Robert, and Prewitt operators, it can provide more precise results [92]. Thus, it is considered to be the most successful and widely used gray edge detection method. In addition, a previous study has shown that the combination of the Canny algorithm and Otsu received less influence from the background value [93]. Figure 8 indicates the superior performance of Canny combined with Otsu.

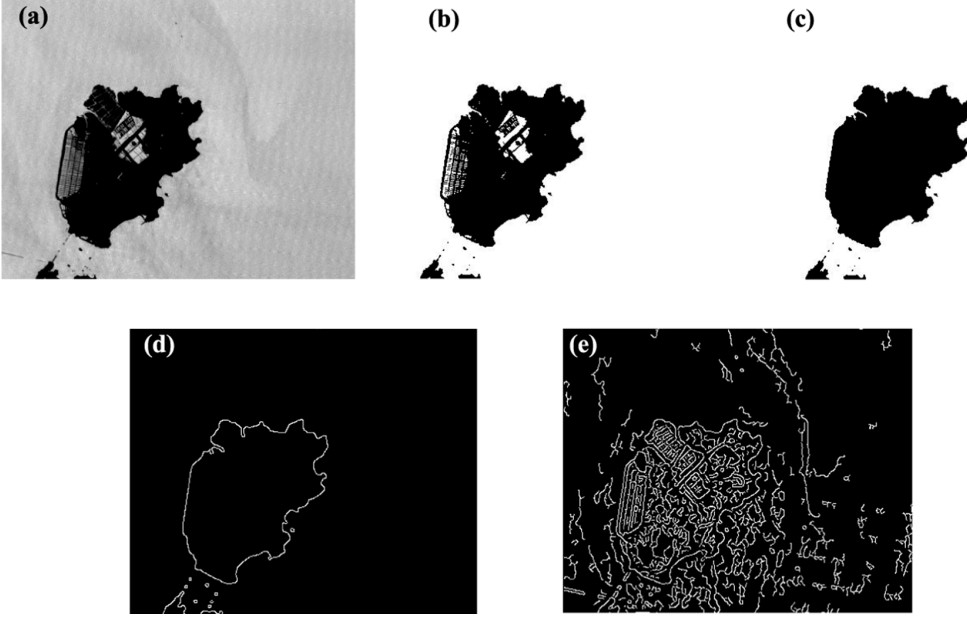

**Figure 8.** Comparison of different extraction methods: (**a**) MNDWI, (**b**) OTSU after MNDWI, (**c**) morphological processes after OTSU, (**d**) Canny edge detection combined with OTSU and morphological processes, and (**e**) Canny edge detection without OTSU [93].

Edge detection methods have the advantages of a simple model and convenient calculation. Among them, the Canny operator has the best performance, with a high signal-to-noise ratio, high positioning accuracy, and strong single-edge response ability, and the detected edge is also clear, delicate, and continuous. However, when the coastline image background is relatively complex, they are usually sensitive to noise and the extraction results are discontinuous, which is not suitable for large-scale coastline extraction. The edge detection method requires manual intervention, and mathematical morphology is used to optimize the results. Accuracy can also be improved by combining with other methods.

### 5.4. Active Contour Model

The active contour model transforms the problem of image segmentation into a problem of solving the minimum of the energy functional to detect the edge of the target [63].

Snakes has gained widespread application in recent years for coastline extraction due to its incorporation of edge smoothness and elastic constraints [94]. The researcher used three models related to the Snakes algorithm based on an up-to-date Landsat mosaic to align with the majority of the Antarctic coastline [95]. Generally speaking, this algorithm is highly effective in extracting coastlines; however, the adjustment of relevant parameters poses a significant challenge and can directly impact the extraction results.

The level set algorithm is a numerical technique that enables the tracking of interface evolution. Since its inception, this method has emerged as an effective approach widely employed in edge detection and contour extraction within the image processing domain [89]. A researcher has proposed two enhanced level set-based algorithms for

extracting coastlines from SAR images [96]. The level set is a collection of points that share the same function value, serving as a representation of a planar curve [97]. Compared to Snakes, the control over the initial contour is automatically adjusted to align with the coastline of the image, without imposing strict constraints [96,98]. The algorithm, however, does possess certain limitations. Specifically, the conventional level set algorithm exhibits sluggishness, particularly when dealing with high-resolution images. Additionally, the evolution process of the level set function often manifests irregularities that can potentially give rise to numerical errors [99]. The researchers proposed a novel distance regularized level set evolutionary algorithm (DRLSE) to address the issue of low accuracy results caused by inadequate initialization of the level set function in coastline extraction [97,99]. Figure 9 shows the results of contour extraction using DRLSE.

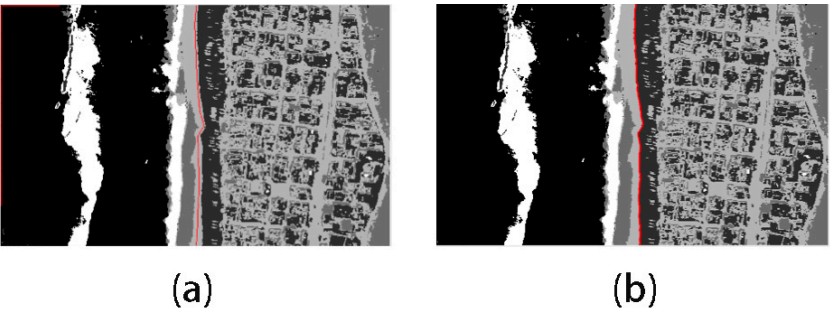

**Figure 9.** The process of approaching the contour line to the edge line in the coastline extraction task [99]. (**a**) shows initial contour with initial values; (**b**) shows contour with final values.

The essence of the active contour model is a variational solution idea. In the process of coastline extraction, we can obtain accurate coastline locations, but the model is complex, and the calculation time is long. This method is suitable for a scene with a simple background and a single object, and the effect is better for high spatial resolution images. For complex background and multi-object scenarios, the model needs to be improved, especially the calculation speed.

*5.5. Polarization Method*

The polarization method is specific to the work of extracting coastlines based on SAR data. In the polarization system of SAR data, the ground information is contained in the electromagnetic wave back from the target object, and this information can be modulated in the components of the spectrum, intensity, or polarization direction of the electromagnetic wave. There are two types of polarization: horizontal polarization (H) and vertical polarization (V). H means that when a satellite sends a signal to the ground, its radio waves vibrate horizontally. V is the opposite. The polarization scattering matrix is a simple method to represent the scattering characteristics of a single pixel, which contains all the polarization information of the target. The scattering matrix of the target is measured by four types of electromagnetic wave polarization such as HH, HV, VH, and VV to determine the nature of the target [100].

Researcher Nunziata from the University of Napoli Parthenope has conducted a series of studies on extracting coastlines from SAR images using polarization methods [64,101–103]. The research team used CSK SAR data obtained by the incoherent double-polarization Ping-Pong mode to achieve coastline extraction [103]. Mimicking the correlation between HH and VV polarization channels was related to the coherence time of the observation scene, leading to the derivation of a binary output that served as the foundation for coastline extraction [101]. Moreover, he proposed a novel metric for dual-polarization coherent and incoherent synthetic aperture radar (SAR) data processing based on the correlation between co-polarization and cross-polarization channels, aiming at coastline extraction as well as unsupervised separation of land and ocean [102]. Except for the study of co-polarization, Paes extracted continuous coastlines using Hybrid Polarity SAR data [104]. In the follow-up

study, in addition to the dual polarization, the research team also used the fully polarized SAR measurement to extract the coastline [105].

In the field of coastline extraction, polarization methods can be classified into two types: single-polarization and multi-polarization. Both methods are effective, but the single-polarization method requires manual refinement when dealing with low land–sea ratios or complex extraction areas. In contrast, the multi-polarization method does not require manual participation and has a faster processing time than the single-polarization method. However, the performance of both methods will decline when the sea level rises to the intertidal zone.

### 5.6. Machine Learning

Machine learning-based approaches for coastline extraction have demonstrated superior performance compared to others. Machine learning methods are highly effective in analyzing big data from remote sensing tasks, as they possess the capability to automatically learn the intricate relationships between input features and output results through extensive calculations [25]. These methods can be well applied to the distinction between sea and beach, bedrock and mangrove, etc., especially in the context of large time scale and large spatial scale, when many remote sensing images need to be calculated. The workflow of using the combination of remote sensing data and machine learning methods for coastline map is presented in Figure 10.

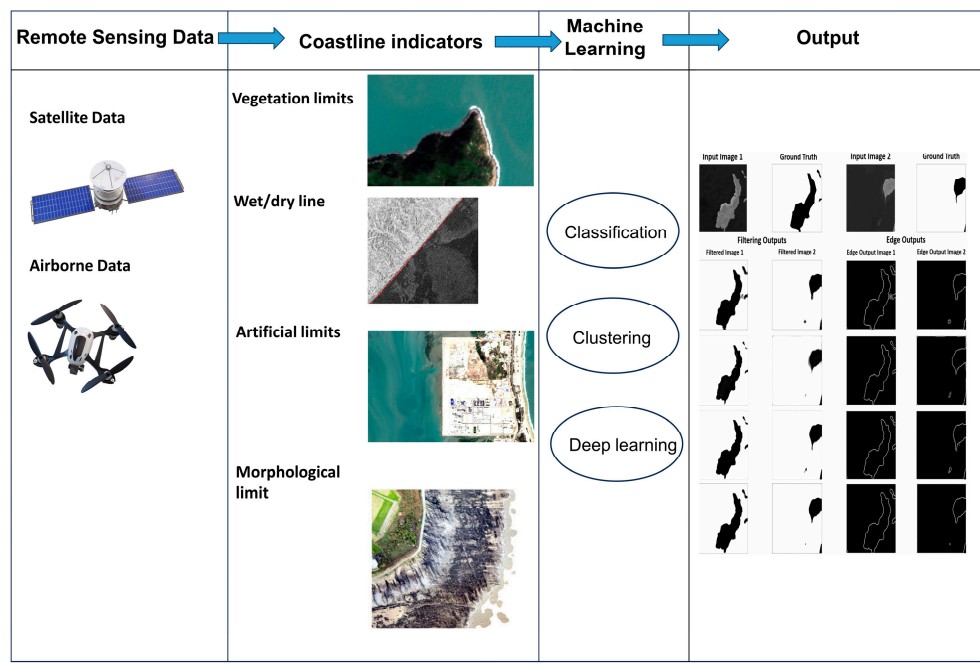

**Figure 10.** Workflow of using the combination of remote sensing data and machine learning methods for extracting coastline.

Generally, machine learning methods can be broadly classified into two primary categories: unsupervised and supervised learning [24]. Supervised learning is a process of using a set of samples from a known class to adjust the parameters of a classifier to achieve the desired performance [55]. In other words, supervised learning trains the existing training samples to obtain the optimal model [106], which can be classified as unknown data.

In contrast to supervised learning, unsupervised learning methods can classify data without the need for target labels. Unsupervised learning includes clustering and dimensionality reduction. Cluster analysis is the process of classifying a sample set based on the similarity between samples, calculated by continuously reducing the intra-class gap and increasing the inter-class gap [107]. Clustering methods are commonly applied for image

processing, landscape object identification, and classification. The primary application of dimensionality reduction algorithms lies in data compression, aiming to mitigate problem complexity, enhance data quality, and ultimately expedite model training.

Deep learning is the most important branch of machine learning and a mainstream method to classify remote sensing objects [108]. By learning the internal rules and representation levels of the sample data, and using the information obtained in the learning process to interpret the data, the ultimate goal of remote sensing image analysis is to enable the machine to analyze, learn, and interpret the image independently [90].

### 5.6.1. Classification

This section presents machine learning methodologies discussed in the existing literature for coastline extraction, wherein the primary approach involves landscape surface type classification to determine precise coastal locations.

The support vector machine (SVM) is a supervised machine learning method commonly employed for classification and regression tasks [109]. The SVM learning strategy involves interval maximization, which can be mathematically formulated as solving a convex quadratic programming problem equivalent to minimizing the regularized hinge loss function. SVM simplifies the classification process by projecting the dataset onto a lower-dimensional feature space. This approach has been extensively utilized in land cover classification, particularly in binary categorization of land and water. Results showed that the SVM classifier is capable of detecting coastline features with sub-pixel accuracy [109]. In addition, a series of algorithms derived from SVM-based algorithms such as SVM-linear (SVM-L), SVM-RBF (SVM-R), and SVM-polynomial (SVM-P) with different kernel function compositions have been compared [110]. Among them, SVM-R is the most effective method, attaining high overall accuracy.

The K-nearest Neighbor (KNN) algorithm is a method used to determine the similarity between a given sample and K samples in the dataset. If the majority of these K samples belong to a specific category, then it can be inferred that the given sample also belongs to that category [111,112]. In this method, unknown pixels can be labeled by examining training pixels. Researchers pointed out that this method can effectively identify the boundary between classes [40]. However, it also can be confused by a zone of transition.

The Parallelepiped algorithm is based on the probability distribution value, utilizing multiple standard deviations as the confidence classification boundary, while the magnitude of the t-value determines the proportional size of each class range. The advantage lies in the simplicity of the classification criteria and computation speed. However, it is found that the results of the parallelepiped algorithm are not accurate, and the extracted results are farther from the reference waterline than other test methods [113].

The Maximum Likelihood is a nonlinear classification method based on the Bayesian criterion with a minimum probability of classification error, which is a widely used and mature supervised classification method [114]. In coastline extraction, Maximum Likelihood performs similarly to the Parallelepiped algorithm. However, the results are still influenced by the wet–dry mixing zone.

The Minimum Distance classification is performed by assuming that the spectral information of each class of features in the image has a multivariate orthogonal distribution, and each class forms an ellipsoidal group of points in the K-dimensional spectral space, and its attribution is determined by the distance of the pixel from the center of each class [115].

The Mahalanobis distance classifier is a weighted Euclidean distance that takes variable correlations into account through the covariance matrix. This technique is similar to the minimum distance. Compared with other methods, this method is unaffected by the magnitude and can exclude the interference of correlation between variables. However, the exaggeration of the effect of variables that exhibit minimal changes leads to a compromise in classification accuracy.

Ensemble learning (EL) classifiers are machine learning methods that employ a series of learners to acquire and utilize rules for integrating the individual learners' learning

outcomes, thereby achieving superior performance compared to a single learner [116]. It is not a single machine learning algorithm per se, but rather a way to build and combine multiple machine learners to accomplish the learning task and to realize the strengths of all. Comparative analysis experiments show that the EL classifier is suitable for beach-type coast extraction, and the extraction accuracy is better than SVM, MLP classifiers, and other algorithms [110]. When there is no strong dependency between individual learners, a series of individual learners can be generated in parallel, and the representative algorithms are the Bagging and Random Forest (RF) algorithms [117]. To extract the coastline, the RF method has been applied to the NIR bands of LANDSAT-8 and GOKTURK-2 images [118]. It can provide pixel-based results. Extra Trees (ET), Adaptive Boosting (AdaB), and Gradient Boosting (GB) were also selected to be applied in coastline extraction [119].

### 5.6.2. Clustering

Clustering is different from classification in that it means dividing similar data together, the specific division does not care about the label of the class, the goal is to aggregate similar data together, and clustering is an unsupervised learning method. The clustering method is divided into three categories, including the partitioning method and the hierarchical method [115]. The partitioning method requires specifying the number of cluster classes or cluster centers in advance and iterating until the final goal of "points within classes are close enough and points between classes are far enough" is achieved. The classical K-Means algorithm is often used for water and land separation. Applying Hierarchical Clustering and Fuzzy C-Means (FCM) extends to water and non-water classification problems, encompassing patch analysis in remote sensing imagery [108].

The K-means algorithm is provided by the function k-means. This function enables the provision of either the cluster center's location or the number of clusters through the center parameter, facilitating multiple randomly initiated partition choices, and ultimately returning the partition with the optimal objective function (minimum sum of squared distances) [120]. However, it should be noted that this does not guarantee an optimal final partition. In the past few years, the K-Means algorithm has found extensive applications in various classification tasks due to its ability to extract features at a deep level effectively. However, there are two major drawbacks in the K-Means algorithm: (1) the K values are pre-selected and fixed and (2) random seed selection may have an impact on the results. K-means++ has been proposed to compensate for these deficiencies when applied to coastline extraction.

The ISODATA (Iterative Self-Organizing Data Analysis) model differs from the K-means mean algorithm in two ways: k-means is a sample-by-sample correction, while the ISODATA statistically examines clusters after each iteration [113]. Additionally, the ISODATA algorithm can not only complete the clustering analysis by adjusting the classes to which the samples belong but also automatically "merge" and "split" the classes to obtain a more reasonable number of classes for each cluster. Several studies about Coast Science used ISODATA to identify characteristics [121]. These studies showed that ISODATA has achieved some success as a built-in algorithm embedded in automated coastline extraction software, where features are classified and merged into two clusters of land and water, and the boundary between the two is used as the coastline [122,123].

The Fuzzy C-Means (FCM) algorithm divides objects based on their similarity, which uses Euclidean space to determine the geometric closeness of the data points as a reference. Researchers compared the application of FCM to K-Means and hierarchical clustering in water extraction, and the results showed that the clustering effect of FCM and K-Means is superior to hierarchical clustering [108].

### 5.6.3. Deep Learning

There needs to be more research applying deep learning techniques to processing remote sensing data for coastline extraction. Previous research has not only compared deep learning methods with traditional approaches but also examined the performance

of different deep learning models. These experiments have demonstrated that machine learning-based coastline detection algorithms have recently begun to outperform traditional statistical methods [124]. In contrast to traditional ML methods, or neural networks, deep learning feature extraction does not rely on manual, but automatic machine extraction. Meanwhile, it provides a comprehensive solution rather than decomposing the problem into distinct components and subsequently integrating them together as in the field of machine learning [72].

The Artificial Neural Network (ANN) is a parallel interconnected system consisting of simple units, which exchange signals through neuron networks and undergo training by initially assigning random values [125,126]. The researchers employed artificial neural network (ANN) models with diverse activation functions to extract distinct categories of coastlines from remote sensing images, and subsequently conducted a comparative analysis with decision trees (DT), k-nearest neighbors (KNN), and support vector machines (SVM) [110]. They mentioned that ANN is an efficient classifier over DT, KNN, and SVM.

The Convolutional Neural Network (CNN) is a feedforward neural network comprising artificial neurons, consisting of a convolutional layer, pooling layer, and fully connected layer [127,128]. The coastline extraction process based on CNN model is shown in Figure 11, which is divided into two parts: training and testing. We need to pre-train the convolutional neural network model and then input the remote sensing image into the trained model for coastline extraction [124]. Various CNN models are put forward with their advantages for different datasets and tasks. Basic CNN models have flourished in water–land segmentation [89]. However, some issues cannot be overlooked, including (1) blurring boundary pixels between water and land and that (2) a large number of trainable parameters and training samples are required [129]. To overcome drawbacks, researchers proposed more robustness and generality-modified models based on CNN architecture [130]. LaeNet is a novel end-to-end lightweight multi-task CNN for automatically extracting lake areas and coastlines from Landsat-8 images, consistent with mainstream semantic segmentation models (UNet, DeepLabV3+, etc.) and even better [131]. To address the challenge of training sample size for network training, researchers propose a deep multi-feature learning architecture called W-net for water body segmentation in satellite images [129]. The problem of blurring boundary pixels in CNN is addressed by proposing a novel loss function, namely edge-weighted loss, for training the segmentation network [130]. Additionally, research on modified CNN networks for separating terrestrial water bodies in complex environments is also under development [106–108]. Despite CNN's commendable achievements in land and water segmentation in recent years, there is still room for further improvement.

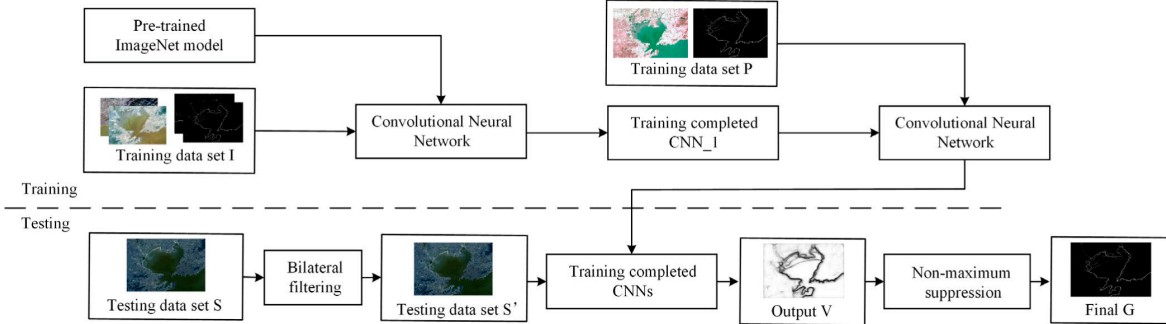

**Figure 11.** Coastline extraction process based on CNN model [130].

The Multi-Layer Perceptron (MLP) consists of an input, output, and multiple hidden layers. The MLP classifier will have a good recognition rate and is faster in classification. In exchange, its training speed is slow and the number of training parameters increases dramatically, especially as the image size increases. The MLPs are combined with water body indices to train the MLPs to distinguish water and land features based on the pixel values of the input satellite image [125]. Additionally, MLP-ANNs can enhance the accuracy of water and land classification. Compared to other classifiers, MLPs not only enable beach

and artificial coastline recognition but also facilitate multiple applications in Markov models for simulating various types of land use change simultaneously, which have been extensively employed in bio-coastline monitoring.

The U-Net is a deep learning network based on the Full Convolutional Network (FCN), widely employed in image segmentation. The model is designed with an encoder-decoder structure, as well as skip connections [132]; the structure is showed in Figure 12. Many neural network models for coastline extraction adopt the fundamental concept of U-Net while introducing novel modules. For instance, researchers modified the Standard U-Net (SUN) model to improve the SLS performance and develop an automatic coastline extraction framework, which overperformed AWEI in results [68]. Researchers proposed models trained using a new Sobel-edge loss function to improve sensitivity to fine-scale, narrow coastline features [66]. Other experiments used multiple U-Net-based models to compare with the integrated automatic coastline segmentation system (WaterNet) [72]. In addition, to build a new network based on conventional U-Net, researchers combined U-Net with the Edge Detection Framework (HED) to solve the duality of coastline extraction tasks (segmentation and representation) [133].

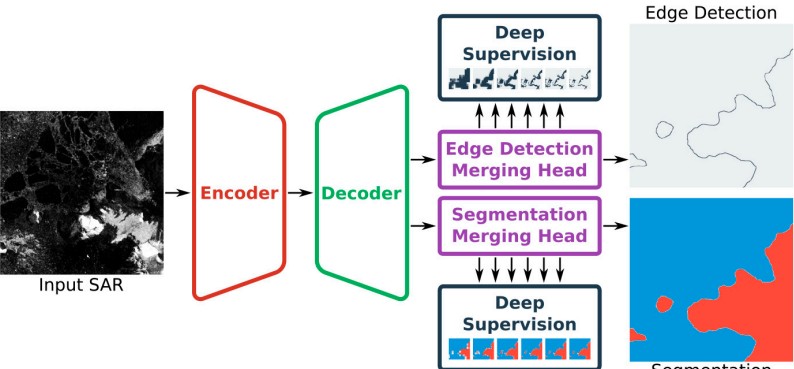

**Figure 12.** The structure of U-NET for extracting coastlines. The encoder and the decoder are used to calculate a pyramid of feature maps [133].

Owing to the DL model training needing substantial data, we have extensively searched publicly available coastal condition datasets. There are five datasets we found, briefly described as follows:

(1)    Sentinel-2 Water Edges Dataset (SWED). This dataset contains 26,468 Sentinel-2 Level-1C images of size $256 \times 256$ and the corresponding sea–land segmented labels. Images annotations were created using a semi-supervised clustering procedure, followed by a manual verification and correction of mislabeled pixels [70].

(2)    Sea–land segmentation benchmark dataset. It contains 3361 Landsat-8 OLI images of size $512 \times 512$. The satellite images were manually labelled by dividing all their pixels into two classes: sea and land [134].

(3)    YTU-WaterNet.  This dataset was also built based on Landsat-8 OLI data, using OpenStreetMap (OSM) water polygon data to generate binary segmentation labels [72]. The final dataset contains 1008 images with a size of $512 \times 512$.

In general, the machine learning method can realize the automatic coastline extraction, but it needs to combine the remote sensing indices and thresholding to improve the accuracy. Deep learning also shows superior performance in coastline extraction but compared with the application of deep learning in image segmentation, object detection, and other image interpretation tasks, the research is still relatively shallow, mainly because there are few public datasets for coastline extraction. However, deep learning for shoreline extraction will still be the trend.

*5.7. Knowledge Graph*

Despite the revolutionary impact of deep learning on remote sensing image classification, current deep learning-based methods exhibit a strong reliance on extensive training data and demonstrate limited performance when confronted with novel categories beyond predefined ones [27]. To solve this problem, DL and Knowledge Graph (KG) fusion is introduced into remote sensing image classification [135]. Among them, KG has the prior knowledge to obtain remote sensing samples for DL. Based on previous studies [25], this paper presents a simple concept for constructing a coastline scene knowledge graph (CSKG).

Accurately describing objects' attributes and spatial relationships can significantly enhance deep learning methods' capacity to comprehend the semantics of remote sensing scenes [25]. Attribute relations can be employed to depict the reflective properties of objects or the inclusive relationships among them. On the other hand, spatial relations primarily delineate diverse positional connections between distinct objects in space [27]. This can be used as a basis for the knowledge-based classification of remote sensing scenes. The construction of a remote sensing knowledge graph (RSKG) shown in Figure 13.

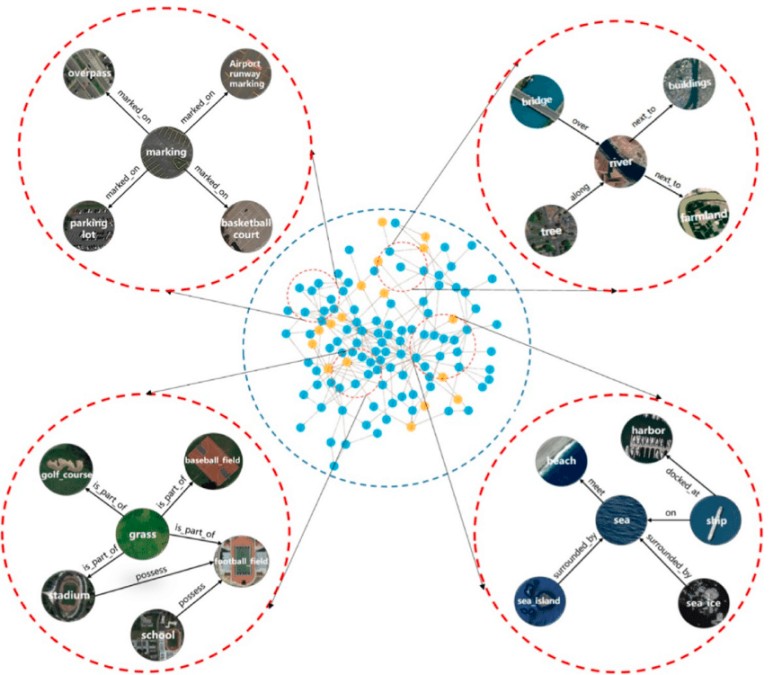

**Figure 13.** The construction of a remote sensing knowledge graph (RSKG) [25].

By consulting several experts in related fields, we make the framework of the knowledge graph. More specifically, we analyze the attribute and spatial relationships of objects in remote sensing (RS) images belonging to specific categories of coastlines in order to construct knowledge graphs (KGs) for distinct types of coastlines shown in Figure 14.

A knowledge graph has the advantages of a low sample data requirement and high computational efficiency. By introducing remote sensing scene classification and combining it with zero-sample learning, knowledge graphs can deal with the problem that deep learning requires a large amount of training data to solve. However, the construction of knowledge graphs requires a lot of expert knowledge, as well as the ability to extract the relationship from the knowledge, which puts forward new ability requirements for coastline management personnel. At present, there is no thematic knowledge map fully used for coastline extraction, which will be another direction for future work.

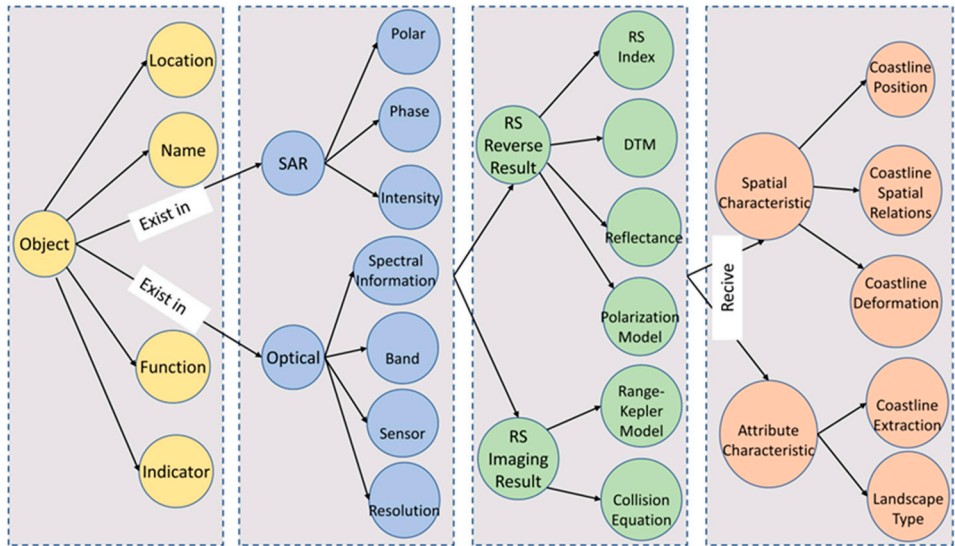

**Figure 14.** The construction of a coastline scene knowledge graph (CSKG).

## 6. Discussions: Challenges and Future Prospective in Coastline Extraction

Coastline extraction is an essential aspect of coastline management, encompassing a series of systematic workflows aimed at determining the precise location of the coastline with optimal efficiency and expediency, facilitating dynamic monitoring. The extraction of coastlines necessitates a sound understanding of fundamental coastal typologies to ascertain more suitable methods for identification and extraction. Depending on the study area's location, artificial infrastructure, vegetation, and other characteristics, the indicators, data types, and coastline extraction methods will differ. Conventional methods for mapping coastlines are constrained by field conditions and limited data resolution, leading to inefficiency and reduced accuracy. As the demand for monitoring coastal environments has grown, researchers have increasingly turned to various data sources and methods to pinpoint coastlines.

With the advancement of remote sensing sensors, the enhancement of computer performance, and the integration of machine learning techniques, real-time waterline recognition, automatic coastline extraction, and real-time monitoring have witnessed significant advancements. Simultaneously, the precise requirements for coastline management, spectral, and multi-temporal remote sensing big data analysis, as well as complex machine learning methodologies, pose additional challenges:

(1) Construction of datasets. By leveraging machine learning methods, more valuable information can be extracted from big data. The introduction of deep learning improves the accuracy of the coastline binary classification problem, and the use of intelligent methods can achieve the fully automated extraction of coastlines. However, the absence of datasets can directly affect the effectiveness of these methods. Creating datasets can supply ample training and testing data for deep learning methods. At present, multiple datasets are available for remote sensing image classification and target recognition, but none of them can be directly applied to coastline extraction. Therefore, there is a need to establish standard coastline image datasets and promote the development of remote sensing image-based coastline extraction technology. The construction of coastline datasets requires two kinds of basic data: coastline data and remote sensing image. Coastline data can be obtained from scientific programs created by government agencies such as the National Oceanic and Atmospheric Administration. Remote sensing imageries are determined based on the date and location of selected stretches of coastline, with Sentinel 2 and Landsat8 currently in common use. The coastline data are combined with the water detection results on the remote sensing image to obtain the sea–land segmentation label, and finally the coastline is extracted

with smaller label images. Coastline data can also be selected using Google Earth images or OpenStreeMap to match remote sensing images. Labeling methods include manual labeling and automatic labeling. Researchers can select the labeling method based on their requirements. When constructing the dataset, it can be considered to construct the dataset that can represent the characteristics of the coastline based on the processed data products such as the index image.

(2)    Select data with appropriate spatial resolution. Choosing a suitable spatial resolution is very important, considering the visibility and easy identification of the coastline indicator. When the eroded coastline size is smaller than the spatial resolution of the image, it is disregarded, leading to an inaccurate coastline extraction. Increasing the spatial resolution enhances indicator visibility but does not necessarily improve classification accuracy. For instance, in the case of coastline indicators such as vegetation lines, spectral characteristics effectively discriminate between vegetation and other features, while higher spatial resolution facilitates the precise extraction of coastlines. However, when the waterline is employed as an indicator, the combination of spectral similarity and extremely high spatial resolution can compromise extraction accuracy. Hence, selecting an appropriate spatial resolution based on specific scenarios during coastline extraction tasks is crucial.

(3)    Using hyperspectral data. The spectral characteristics of offshore areas exhibit a high degree of complexity, posing challenges in distinguishing the diverse features of coastal zones using single spectral information in conventional methods. However, with advancements in remote sensing sensors and the increasing availability of hyperspectral data, a novel approach emerges for extracting multi-band and abundant spectral information from coastlines. This enables the fusion of multiple bands' spectral information to accurately delineate the distinctive characteristics of coastal zones.

(4)    Coping with the effects of the seasons and weather. In order to extract reliable indicators of coastline dynamics, it is essential to consider seasonal variations. The composition of many beaches undergoes changes throughout the seasons, such as shifts from sand to gravel and fluctuations in vegetation growth or withering. These alterations not only impact the visibility and accuracy of coastline indicators but also influence the slope of the coastal zone. Consequently, a uniform method for coastline extraction may not be suitable across different seasons. Therefore, we argue that conducting seasonal phase-based studies on coastline extraction and erosion would yield more valuable insights. Additionally, during periods of strong winds leading to large waves at sea, interference with transient waterline detection becomes particularly prominent in SAR data-based coastline extraction, thus, weather conditions and climate factors should be considered as they can significantly affect the outcome of coastline extraction efforts.

(5)    Improve the availability of SAR data. The separation of land and sea and the extraction of coastline using SAR data are highly feasible; however, due to the noisy nature of the data and difficulties in preprocessing, decoding accuracy is often low. The strong backscattering caused by wind and wave modulation on the sea surface greatly reduces contrast between land and sea, resulting in weak boundaries that are difficult to extract.

(6)    Combine various methods. From the analysis of this whole paper, it can be seen that various methods for the automatic extraction of coastlines from remote sensing images compiled in this paper have certain limitations, and each method is only for a specific coastline type, lacking universality. Therefore, in subsequent work, we can consider combining various methods and integrating the advantages of each method to improve the coastline extraction effect.

(7)    Realize coastline extraction with sub-pixel accuracy. The current coastline extraction experiments show that pixel-level extraction accuracy can be achieved, i.e., classification of each pixel. In fact, due to the transitional and variable nature of the coastline, the same pixel can be partially classified as a seawater region and partially classi-

fied as a land region. In data applications with low spatial resolution, especially for multispectral as well as hyperspectral data, there is a very typical phenomenon of mixed pixels, and it is necessary to perform pixels such that coastline extraction can be achieved at sub-pixel level accuracy and more accurate monitoring of coastlines. At the same time, the complex microtopography of the coastline can be realized.

(8) Construction of remote sensing knowledge graphs. We will try to extend CSKG to the remote sensing scene knowledge graph (RSKG). Incorporating a larger repository of relevant structured knowledge graphs (RSKG) can enhance the availability of comprehensive prior knowledge, thereby facilitating the generation of more refined semantic representations for both RS scenarios and target classes. Monitoring of coastline changes can be achieved from a single-state remote sensing ontology structure to a sequential state ontology structure.

## 7. Conclusions

With the rapid development of coastal economic belts, the significance of real-time monitoring of coastline changes is increasingly prominent, and it is crucial to accurately and efficiently measure and process coastline data [83]. This paper provides a concise overview of the research progress in remote sensing-based coastline extraction, encompassing data sources, types of coastlines, indicators, and algorithm models. Additionally, we highlight existing challenges while proposing potential solutions.

Introducing remote sensing big data can realize the extraction and temporal detection of large-scale coastlines. Firstly, we provide a comprehensive overview of the utilization of remote sensing data in coastal research, encompassing both satellite data and non-satellite data. Specifically, we carefully summarize the sensor parameters used for shoreline extraction. We also compare the advantages and disadvantages of different data for different shoreline types and recommend the most suitable data for different types of shoreline extraction work, which helps provide researchers with valuable insights intuitively. Subsequently, we present an exhaustive compilation of mainstream coastline extraction methods including RS index, thresholding, edge detection, polarization method, and machine learning, the data of which are from the coastline survey using the RS system. We summarized the usage, advantages, limitations, and possible directions of these approaches.

Machine learning has been widely applied as a data-driven tool in remote sensing data processing [133]. We used some length to summarize the machine learning methods for coastline extraction, which can be categorized into three main groups: classification, clustering, and deep learning. Advanced deep learning approaches enable the processing of vast amounts of high-dimensional remote sensing data with enhanced spectral and temporal features, thereby enhancing the accuracy of extraction and monitoring. In addition, based on RSKG, we tentatively put forward the framework of CSKG, which opens up a new idea for subsequent coastline management.

Coastline monitoring is an important link in coastal zone resource management, sustainable development, and ecological protection. It is very important to use advanced and accurate methods to extract coastline for the sustainable monitoring and development of coastlines. The technologies summarized in this paper and the development trends proposed in this paper provide a direction for the procedural and practical application of coastline extraction, help to promote the engineering of coastline extraction, and help the government and Marine ecologists to limit the negative impact of overexploitation of coastlines on the ecology, environment, and climate.

**Author Contributions:** Collect and sort out literature, X.Z. and F.Z.; writing—original draft preparation, X.Z. and J.W.; writing—review and editing, X.Z., H.Y. and H.W.; proofreading, X.Z., J.W., F.Z., H.W. and H.Y. All authors have read and agreed to the published version of the manuscript.

**Funding:** This research received no external funding.

**Data Availability Statement:** Not applicable.

**Acknowledgments:** We would like to thank the anonymous reviewers for their insightful comments and suggestions.

**Conflicts of Interest:** The authors declare no conflict of interest.

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
