# Peer review of "An Overview of Coastline Extraction from Remote Sensing Data"

_remotesensing, doi:10.3390/rs15194865_

Round 1

Reviewer 1 Report

This paper provides a review in coastal researches using remote sensing data and technologies. Some key points such as data sources, extraction methods, remote sensing-based index are summarized in a concise way. The following issues should be addressed:

(1)    Only few keywords were used for literature searching. Some literatures might be omitted. For instance, Sci Data 8, 304 (2021), RSE 278 (2022) 1130, etc.

(2)    As connected to Table 1, an illustration consisting of photos and corresponding remotely sensed images for different coastline types is likely desirable.

(3)    A feature summary regarding remote sensing products of global-scale coastal conditions (e.g. RSE 278 (2022) 1130) is needed. Data source, resolution, methods and other related information should be provided.

(4)    The use of each method and its advantages, especially where improvements need to be made, are expected to be further elaborated.

(5)    It is suggested that a diagram be added to summarize the framework of this article and the methods included。

(6)    The text in the Figure 1 is too small and this figure needs to be improved.

Author Response

Dear reviewer,
   Please see the attachment.  We have provided point-by-point response to your comments in our attachment. Thank you very much for your time involved in reviewing the manuscript and your very encouraging comments on the merits.

Sincerely,

The Authors

Reviewer 2 Report

the paper is well-written and structured also the work is interesting. I suggest to your author to update the literature review and introduction section with the recent publication. read this paper for more details 

  • DOI: 
  • 10.1109/ACCESS.2020.3027881  

check the paper against the journal template

future work and limitations are suggested 

what are the benefits of this work for industrial

the content of the paper should change according to the scientific paper to rise the quality of the paper 

need a revision

Author Response

(The authors gave the same response as above.)

Reviewer 3 Report

Coastline extraction cannot be separated from natural processes, if this is not taken into account it could lead to misleading (Susilowati et al Regional Studies in Marine Sciences, 52. 2022). This paper reviews various current methods of coastline extraction using remote sensing technology. Although the novelty of this paper is not large it is very useful for other researchers who use remote sensing as a tool for their research on coastlines.

It is better to add a paragraph on the difficulties and advantages between optical satellites or radars with different types of coasts. For example, optical imagery is highly recommended for muddy beaches etc. This paper needs to display the results of image processing to provide an overview to its readers, not just to collect the results of literature studies.

Author Response

(The authors gave the same response as above.)

Reviewer 4 Report

Dear authors,

I have read your study carefully. It is a good review. As you can see in the attached PDF, I marked the necessary clarifications  and corrections, and I also corrected some sentences (as suggestions)

There were some problems with English; I tried to fix what I saw. Please check.

Author Response

Dear reviewer,

Thank you very much for your time involved in reviewing the manuscript and your very encouraging comments on the merits.

“I have read your study carefully. It is a good review.”

We also appreciate your clear and detailed feedback. We have carefully read the file you uploaded, and we are very moved by your serious attitude.At the same time, we are very sorry for the trouble caused to you by our poor English.

We tried our best to improve the manuscript and made some changes to the manuscript.  These changes will not influence the content and framework of the paper.  And here we did not list the changes but marked in red in the revised paper. We appreciate for your warm work earnestly and hope that the correction will meet with approval.

We would like to take this opportunity to thank you for all your time involved and this great opportunity for us to improve the manuscript. We hope you will find this revised version satisfactory.

Sincerely,

The Authors

Reviewer 5 Report

Dear authors,

The paper provides a comprehensive overview of coastline extraction methodologies using remote sensing techniques. It discusses various data sources, including satellite and non-satellite data, and highlights the importance of accurate coastline delineation for coastal zone management. The paper categorizes coastline indicators and presents a detailed analysis of coastline extraction methods, encompassing traditional approaches, machine learning, and deep learning techniques. Strengths and weaknesses of these methods are discussed, along with the challenges posed by factors like seasonal variations and data resolution. The paper emphasizes the need for standardized coastline image datasets and suggests the integration of knowledge graphs for improved coastline tracking. Overall, it offers valuable insights into the evolving field of coastline extraction and outlines potential avenues for future research and development.

The article, from my viewpoint, is characterized by these strengths that make it special:

·        Comprehensive Overview: The paper provides a comprehensive overview of coastline extraction methods, data sources, and indicators. It covers a wide range of topics related to coastline extraction, making it a valuable resource for researchers in the field.

·        Structured Presentation: The content is well-structured, with clear section headings that make it easy for readers to navigate through the paper. Using subsections within each main section adds to the clarity.

·        Inclusion of Challenges: The paper appropriately identifies and discusses the challenges associated with coastline extraction, which adds depth to the discussion. This helps readers understand the complexities of the field.

·        Integration of Machine Learning: The incorporation of machine learning techniques in coastline extraction is well-documented. The categorization of machine learning methods into classification, clustering, and deep learning is informative and useful for readers.

·        Future Prospects: The paper presents thoughtful suggestions for future research directions, such as sub-pixel accuracy and the use of knowledge graphs. These ideas can guide researchers in advancing the field.

However, from my point of view, I find that the article also has some weaknesses.

1.        Citation and References: While the paper is generally informative, it lacks specific citations for some statements made, especially in the introduction and background sections. It is essential to provide proper references to support claims.

A.      Incomplete Citations: The paper occasionally lacks complete and accurate citations. For instance, in Section 5.2 (S306), it mentions "previous studies" without providing specific references or authors (only reference [57]), making it challenging for readers to verify the claims or locate the cited studies.

B.      Outdated References: Some of the cited references in the paper are notably outdated, with publication dates exceeding a decade ago. This raises concerns about the currency of the sources used to support the research.

C.      Missing Citations: There are instances where key concepts or statements are presented without proper citation. For example, in Section 5.4 (S497), when discussing polarization methods, it introduces "Nunziata" without a prior citation or explanation of who Nunziata is or their work.

D.     Lack of Peer-Reviewed Sources: A significant portion of the cited references appears to be from conference papers or non-peer-reviewed sources. While these sources can be valuable, a more balanced inclusion of peer-reviewed research would enhance the paper's credibility.

E.      Overreliance on Self-Citation: The paper seems to rely heavily on self-citations, which can create a bias and may suggest an attempt to inflate the importance of the authors' previous work.

2.        Data Availability: The paper mentions the importance of constructing datasets for machine learning methods but does not provide practical guidance on how to address this challenge. I recommend including information on potential sources or initiatives for building coastline image datasets would be helpful.

3.        In-Depth Explanation: Some sections, particularly those discussing machine learning methods, could benefit from more detailed explanations and examples. To enhance understanding, it is recommended to solve the following:

A.      Insufficient Background Information: In several sections, the paper lacks adequate background information to help readers understand key concepts. For example, in Section 5.1, when discussing "coastline indicators," the paper assumes prior knowledge and doesn't explain what these indicators are or why they are important.

B.      Lack of Clarification on Methodology: The paper sometimes introduces complex methodologies without providing a detailed explanation. For instance, in Section 5.4, when discussing the "polarization method," it briefly mentions "scattering matrix" and "co-polarization" without elaborating on these concepts, making it difficult for readers unfamiliar with SAR data to follow the discussion.

C.      Unclear Terminology: There are instances where the paper uses technical terminology without clear definitions or explanations. In Section 5.2, it mentions "intertidal zone" without defining this term, potentially leaving readers puzzled about its significance.

D.     Failure to Address Potential Reader Questions: The paper doesn't always anticipate or address questions that readers might have. For instance, in Section 5.3, when discussing "threshold segmentation methods," it doesn't explain why thresholding is a suitable technique or when it might be preferable over other methods.

4.        Conclusion Recap: The conclusion section could summarize the main findings and contributions of the paper more explicitly. It should repeat the key takeaways and emphasize the significance of the research.

I suggest to improve the article, following the previous indications in the following way:

1.      Citation and References: Ensure that all statements and claims are properly supported by relevant citations. This strengthens the credibility of the paper.

2.      Practical Guidance: Offer practical guidance on constructing datasets for coastline extraction. Mention any existing initiatives or resources where researchers can access relevant data.

3.      Illustrative Examples: Incorporate illustrative examples or case studies, particularly in the machine learning section, to help readers grasp the practical application of these methods in coastline extraction.

4.      Conclusion Recap: In the conclusion section, succinctly recap the main findings, contributions, and implications of the paper. Emphasize the importance of the research in advancing coastline extraction techniques.

The paper offers a valuable contribution to the field of coastline extraction. Addressing the weaknesses and starting the suggested improvements would enhance its clarity and impact, making it an even more valuable resource for researchers and practitioners in coastal management and remote sensing.

The English quality of the paper, is generally good, but there are areas where improvement is needed. Please find recommendations and examples of how to enhance it:

Grammar and Syntax:

Example: Original - "The polarimetric co¬registered channels commonly contain HH, VV or HH, HV or VV, and VH[62]."

Suggested Improvement - "Polarimetric co-registered channels typically include combinations of HH, VV, HV, and VH[62]."

Example: Original - "To address the nonlinearity of remote sensing datasets, several studies have indicated that these methods are useful."

Suggested Improvement - "To account for the nonlinearity of remote sensing datasets, several studies have shown the utility of these methods."

Word Choice:

Example: Original - "The machine learning-based coastline detection algorithm has started to outperform conventional statistical methods[124]."

Suggested Improvement - "Machine learning-based coastline detection algorithms have recently begun to outperform traditional statistical methods[124]."

Example: Original - "The establishment of datasets can provide sufficient training data and test data for deep learning methods."

Suggested Improvement - "Creating datasets can supply ample training and testing data for deep learning methods."

Citations and References:

Verify the accuracy and consistency of citation styles throughout the paper, adhering to a specific style guide (e.g., APA).

Complex Language:

Simplify overly complex language in certain sections to make the paper more accessible to a broader audience. For example, break down complex sentences into smaller, more digestible ones.

Example: Original - "In response to the growing demand for coastal zone environmental monitoring and scientific advancement, researchers have increasingly adopted diverse data sources and methodologies to ascertain coastline locations."

Suggested Improvement - "As the demand for monitoring coastal environments has grown, researchers have increasingly turned to various data sources and methods to pinpoint coastlines."

Author Response

Dear reviewer,
   Please see the attachment. We are very sorry that the first version file we uploaded is wrong, please read the second version directly.We have provided point-by-point response to your comments in our attachment. Thank you very much for your time involved in reviewing the manuscript and your very encouraging comments on the merits.

Sincerely,

The Authors

Round 2

Reviewer 5 Report

Dear authors,

Congratulations for the effort to include all the suggestions. The paper now provides a better comprehensive overview of coastline extraction methodologies using remote sensing techniques.

Regards,

The English quality of the paper has improved from the previous version.